# Classifying California's stream thermal regimes for cold-water conservation

**Ann D. Willis**[1☯]*, **Ryan A. Peek**[1☯], **Andrew L. Rypel**[1,2]

**1** Center for Watershed Sciences, University of California, Davis, California, United States of America,
**2** Wildlife, Fish, & Conservation Biology, University of California, Davis, California, United States of America

☯ These authors contributed equally to this work.
* awillis@ucdavis.edu

**Data Availability Statement:** Data and code are available via GitHub repository: https://github.com/ucd-cws/streamtemp_classification.

## Abstract

Stream temperature science and management is rapidly shifting from single-metric driven approaches to multi-metric, thermal regime characterizations of streamscapes. Given considerable investments in recovery of cold-water fisheries (e.g., Pacific salmon and other declining native species), understanding where cold water is likely to persist, and how cold-water thermal regimes vary, is critical for conservation. California's unique position at the southern end of cold-water ecosystems in the northern hemisphere, variable geography and hydrology, and extensive flow regulation requires a systematic approach to thermal regime classification. We used publicly available, long-term (> 8 years) stream temperature data from 77 sites across California to model their thermal regimes, calculate three temperature metrics, and use the metrics to classify each regime with an agglomerative nesting algorithm. Then, we assessed the variation in each class and considered underlying physical or anthropogenic factors that could explain differences between classes. Finally, we considered how different classes might fit existing criteria for cool- or cold-water thermal regimes, and how those differences complicate efforts to manage stream temperature through regulation. Our results demonstrate that cool- and cold-water thermal regimes vary spatially across California. Several salient findings emerge from this study. Groundwater-dominated streams are a ubiquitous, but as yet, poorly explored class of thermal regimes. Further, flow regulation below dams imposes serial discontinuities, including artificial thermal regimes on downstream ecosystems. Finally, and contrary to what is often assumed, California reservoirs do not contain sufficient cold-water storage to replicate desirable, reach-scale thermal regimes. While barriers to cold-water conservation are considerable and the trajectory of cold-water species towards extinction is dire, protecting reaches that demonstrate resilience to climate warming remains worthwhile.

## Introduction

Water temperature influences many biological, physical, and chemical processes in stream ecosystems [1–3]. While some research explores the behavioral response of aquatic organisms across stream temperature thresholds, particularly in a regulatory or management context

**Funding:** AW received funding from the S.D. Bechtel, Jr. Foundation via an unrestricted donation to the U.C. Davis Center for Watershed Sciences and the John Muir Institute for the Environment (fund number 07427) for this work. RP received internal funding from the John Muir Institute for the Environment (fund number 07427). Dr. Jay Lund of the U.C. Davis Center for Watershed Sciences was major advisor to Dr. Ann Willis and provided feedback to the draft manuscript. Dr. Andrew Rypel, co-director of the U.C. Davis Center for Watershed Sciences, co-authored the manuscript and provided advice on study design, data analysis, and preparation of the manuscript. The S.D. Bechtel Jr. Foundation and John Muir Institute of the Environment had no role in study design, data collection and analysis, decision to publish, or preparation of the manuscript. The website for the S.D. Bechtel, Jr. Foundation is http://sdbjrfoundation.org/. The website for the UC Davis Center for Watershed Sciences is watershed. ucdavis.edu. The website for the John Muir Institute of the Environment is johnmuir.ucdavis. edu. The authors confirm there are no real or perceived financial conflicts of interest.

**Competing interests:** The authors have declared that no competing interests exist.

[e.g., 4,5], other work considers annual stream temperature patterns, or thermal regimes, to characterize the dynamics between stream temperature and aquatic ecosystems [2,4]. Analogous to flow regimes, thermal regimes characterize the magnitude, frequency, duration, timing, and rate of change in water temperature [5]. An annual time series of water temperature data defines a thermal regime for a specific location, whereas thermal landscapes consider the pattern of thermal regimes over an entire region [4,6]. Thermal regime research from the refugia- to reach-scale has explored the relationship between the timing, magnitude, and extent of exposure to both elevated and cool water temperatures for limits to and overall productivity of aquatic ecosystems [7–11].

Given overall trends of stream warming due to climate change, the constriction and loss of habitat that supports coldwater species, such as salmonids, is a particular concern in land and water management [8,12–14]. Globally, warming of thermal landscapes are a direct product of climate change [15–17]. Across the United States, projections show nearly 50% of cold-water habitat could be lost due to climate change [8,18], though this decline varies widely depending on species, their thermal constraints, and landscape resistance to dispersal [18]. These changes are compounded by the regulation effects of dams. For example, changes in the timing and magnitude of peak temperatures caused by climate warming and dam regulation in the Columbia River and its tributaries have contributed to declining salmon populations [12,19]. For freshwater aquatic organisms, regulated thermal regimes alter important cues and processes for life-history strategies that evolved in unregulated regimes [7,20–22].

Ecosystem classification frameworks are an important tool for facilitating improved natural resource conservation [23–26]. Classification tools assist in more accurate analysis and comparisons of ecosystems and provide a science-based language for communication with stakeholders and policy makers. For example, given limited resources and the desire to target conservation investments for maximum environmental benefits [27], identifying long-term, viable cold-water habitats is critical [14]. Thermal regime modelling and classification has been widely used to characterize spatial and temporal thermal variability within and across watersheds and regions [28–33]. However, natural resource management agencies in the United States have struggled to integrate concepts of thermal regimes and landscapes into strategies that target species of conservation or economic importance [6].

In California, cold-water conservation is complicated by geography and engineering. California's Mediterranean climate includes extreme climatic and hydrologic variability [34]. At the southern extent of many cold-water fish species in the northern hemisphere, climate warming is likely to shrink the extent of unregulated cold-water habitat [35]. But unregulated reaches account for a small fraction of existing cold-water habitat: over 1,400 dams are on streams relevant to native fish conservation, making available habitat highly regulated [36]. However, despite numerous engineering studies that suggest dams ban be operated to achieve desirable thermal regimes [37–39], few of these studies test those operational hypotheses against the extensive work (that primarily exsts in the ecological research community] around thermal regimes necessary to support aquatic ecosystems [6,20,40–42].

In California, this scientific dissonance combines with its geographic vulnerability to climate change to create a powerful confluence of ambitious ecosystem goals for highly regulated streams with severe consequences if they are not achieved [43,44]. Water management and land use changes have already changed thermal regimes throughout the state, with warmer temperatures reducing the distribution and survival of cold-water fish species [45]. Previous studies of thermal regimes for cold-water ecosystems in California have generally neglected regulated reaches [8,24,31,35], and either explored California as part of a national analysis [8,31] or have focused on a specific region within the state [35,39,40,46]. Where studies have explored the effects of dams, the results suggest that they produce variable thermal regimes

depending on size, the ability to selectively withdraw water depending on temperature (i.e., whether a dam possesses the necessary infrastructure to adjust the depth at which it draws water for releases), and operational objective(s) [47,48]; these regulated thermal regimes may or may not align with existing, unregulated regimes. In addition, methods used for some of these analyses rely on numerical modelling [35,39] or costly data collection [40]: resource-intensive approaches that are impractical for a statewide analysis. Other approaches that require less data (< 5 years) or are computationally efficient bring considerable uncertainty in the results [8,42,49].

This study develops a stream classification framework for California's thermal regimes that allows for rapid identification of stream reaches with cool- and cold-water thermal regimes. In doing so, several fundamental questions related to cold-water conservation are are addressed. First, what constitutes a cold-water thermal regime, and how does it vary across a region? Second, do dams reset the longitudinal evolution of thermal regimes along a streamscape? Finally, can dams be used to manage and replicate desirable cold-water regimes? While this study focuses on cold-water habitat in California, the results can be applied to any region and ecosystem to explore how their thermal regimes may be distinct from alternative locations. The study results can help evaluate which stream reaches should be targeted for cold-water conservation. In addition, these results can be used to assess alternatives, such as whether regulated reaches are suitable trade-offs to historical conditions in unregulated habitat.

## Data and methods

### Data sources and site selection criteria

Stream temperature data were used to model the thermal regime for 77 stream sites throughout California. Data were downloaded from the United States Geological Survey (USGS) and California Data Exchange Center (CDEC), publicly accessible databases. Monitoring sites were initially filtered to exclude those in the Sacramento-San Joaquin Delta region to focus the analysis on freshwater thermal regimes and minimize the influence of tidal dynamics. Recent studies have recommended at least 8 to 12 years of continuous, daily average temperature data to reduce uncertainty from interannual variability [32,33]. Monitoring stations were filtered to identify those with at least 8 years of daily average stream temperature data to balance the desire for reduced uncertainty with sufficient spatial representation. Additional data were used from a long-term (10-year) monitoring network in the Shasta River watershed in Siskiyou County, northern California [46,50,51]. Final sites were located in 7 of 10 hydrologic regions of the state (as defined by the California Department of Water Resources: North Coast, North Lahontan, Sacramento River, San Francisco Bay, San Joaquin River, South Lahontan, Tulare Lake); no sites in the southern range of the state (Central Coast, South Coast, and Colorado River) had sufficient periods of record for our analysis. All data were reviewed to remove flagged data (per USGS and CDEC standards) and obvious outliers; the remaining years with a minimum of 8 daily average temperature observations for each day were used in the study. Any data gaps or missing values were not filled; the data was used as is. Daily average stream temperatures were calculated from sub-daily data sets using R (version 4.0) with RStudio [52,53].

### Thermal regime modelling and classification

Thermal regime modelling used the mosaic package in R [54]. The reviewed datasets were used to calculate annual thermal regimes, defined by the daily mean temperature for each day of the water year (October 1 through September 30). Annual thermal regimes were modelled

with a sine function [29,30]:

$$T_w = a + b\sin\frac{2\pi}{365}(n + n_o)$$

where $Tw$ is water temperature, $n$ is the day of water year, and $a$, $b$, and $n_o$ are coefficients that correspond to annual mean, annual amplitude, and phase (Fig 1). Coefficients $a$, $b$, and $n_o$ were optimized using least square regression. Model fitness for each site was quantified using residual standard error; values closer to zero indicated better fit. Modelled thermal regimes were classified based on clustering and statistical analysis methods developed in Maheu et al. [31], which are briefly summarized here: mean, amplitude (i.e., the difference between the annual mean and annual maximum water temperature), and phase (i.e., day of water year when annual maximum occurs) metrics were calculated from each thermal regime model (Fig 1), then classified using Ward's method, an agglomerative nesting algorithm. Each class comprises a cluster of individual sites, and is defined based on unique features of the clustering parameters (e.g, a cluster of sites in a class all show similar annual variability and amplitudes that are distinct from the other classes). The number of classes was determined using Calinski and Harabasz's (CH) index and the sum of squares ("Elbow") method [55]. In addition, we used the silhouette method to validate the appropriate number of classes [56]. Classes were examined for stability using the Jaccard coefficient, with stable clusters indicated by coefficients greater than 0.75 [57]. Clustering and statistical indices were computed using R packages cluster [58], factoextra [59], and fpc [60].

To further assess the relationship between each thermal metric and a given cluster, we used principal components analysis to describe the variation associated with each metric (annual mean, amplitude, and phase). We visualized the distribution of the clusters with an ordination plot of the first two principal components from the analysis, grouped by cluster. We examined

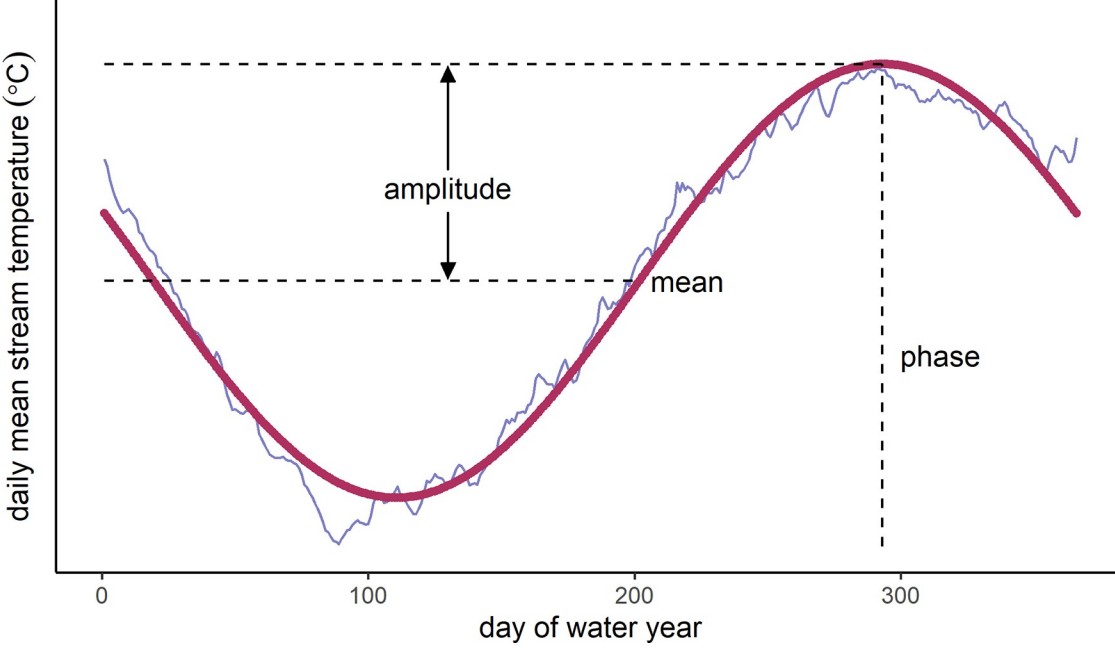

**Fig 1. A thermal regime model fit to observed data.** This example uses data from USGS site 11390500 (Sacramento River below Wilkins Slough near Grimes, CA). Cluster analysis is based on annual mean, amplitude, and phase metrics for each thermal regime model. Amplitude and phase are analogous to annual maximum and day of annual maximum metrics. Figure adapted with permission from Maheu et al. [31]: Fig 1.

relative contributions of each parameter to each principal component to determine which was more important to final clustering results. We also used the Principal Components Analysis (PCA) to identify the centroid of each cluster; then, we calculated the distance of each cluster member relative to the centroid of its respective cluster to identify weak members. For each thermal regime class, a histogram was made to examine the distribution of distance to centroid across all members. Weak members were defined as sites located furthest from the centroid. Additional clustering analysis was done using the same methods to assess whether weak thermal class members had fundamentally different dynamics that were lost in higher-order clustering, or were simply geographically distant from strong members in the same gradient or regime.

### Influence of dam regulation

Dam regulation effects were examined by quantitatively and qualitatively assessing thermal regime patterns downstream of dams. The discontinuity distance downstream of a dam depends on many factors, including dam size, location on a river (e.g., headwaters versus lower reaches), operational objectives, and release strategy (e.g. hypolimnial release; [21]). Previous research showed that large dams in California's Central Valley often influence thermal regimes 30–60 km downstream of release outlets [20,39]; Shasta Dam, impounding California's largest reservoir, was shown to influence temperature patterns up to 250 km downstream [61]. Because varied dam sizes and reaches with multiple dams may show varied effects, histograms were generated for each thermal regime using sites that were within mainstem reaches 100 km or less downstream of a dam. Finally, thermal regimes in these regulated reaches were examined for member strength of each below-dam site to its respective regime class. We compared the distance of each site to its respective upstream dam to its strength as a member to its cluster as quantified by the PCA.

### Results

Modelling results showed a reasonable sine curve fit for all sites included in the study (all model results, including regime classification and calculated metrics, and site metadata are included in S1 Table). Of the 77 sites, 53 had residual standard errors $< 1.0°C$, and all but two had residual standard errors $< 2.0°C$ (Fig 2). Poorer model fits tended to occur at sites with greater temperature variability.

The clustering analysis showed California's thermal regimes were best divided into either three or five classes. An inspection of each result showed that k = 3 produced generally coarse groupings with little insight to the nuances of various thermal regimes. The five-class system was preferred due to its strong coefficients (Ward's agglomerative coefficient = 0.96; CH index ranked k = 5 next favorable behind k = 3), and more refined characterization of the thermal landscape.

The two principal components represent different orthogonal variation of the three parameters (annual mean, annual amplitude, and phase), and together accounted for 88.6% of the total variation in the temperature data ([1–5], Fig 3A). PC1 was most correlated with annual mean temperature (46.6% variance explained) and annual amplitude (46.8%) in a thermal regime; PC2 was most strongly correlated with the effect of phase (93.4%). The five clusters generally represent unique combinations of thermal regime characteristics, with the exception of groups 2 and 4. Groups 1 and 3 were the least variable as expressed by Jaccard coefficients ($J_c$) of 0.86 and 0.95, respectively. Groups 2 and 4 had considerable overlap, and were more variable ($J_c$ = 0.63 and 0.59, respectively); Group 5 was similarly variable ($J_c$ = 0.63). Despite this instability, further examination of results showed a strong physical basis for each grouping. An examination of the elbow and silhouette results also supported k = 5 as the appropriate number of clusters (Fig 3B and 3C).

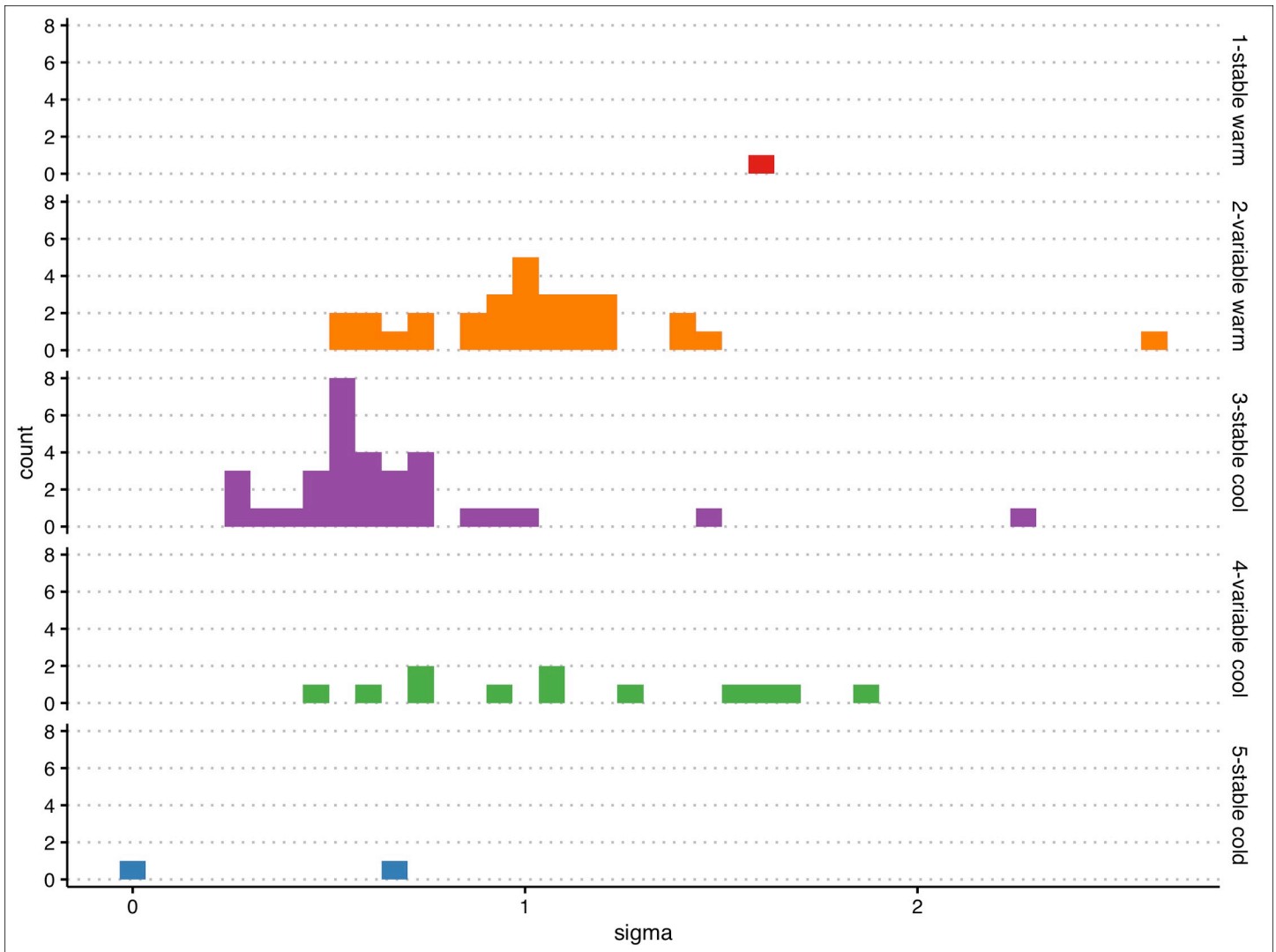

**Fig 2. Histograms of residual standard error, grouped by thermal class.**

Because group 1 had a single member (USGS gage 10265150, Hot Creek in South Lahontan hydrologic region), it was not included in the weak-member analysis. Of group 5's two members, the Shasta Dam outlet (SHD) was almost twice as far from the group's centroid (43.2 units away from the centroid) as the other member, a groundwater-fed spring source (BSC_spring, 22 units; Fig 3). The remaining groups showed tighter membership around their centroids. The cluster plot of group 2 showed that most members were on the perimeter of the cluster, with distances from the centroid ranging 0.5–6.3 units, suggesting a highly variable range of member strength to the cluster. Group 3 was tightly distributed around its centroid (4.0–7.8 units) despite having a larger population (n = 30). Group 4 was slightly more dispersed, with member distances from the centroid ranging 3.3–8.6 units; unlike groups 2 and 3, members were scattered both within the cluster and around the perimeter. When sites were reclassified to allow for additional clusters (k = 6), all groups retained the same membership except group 4, which split into two clusters (n = 20 and n = 12).

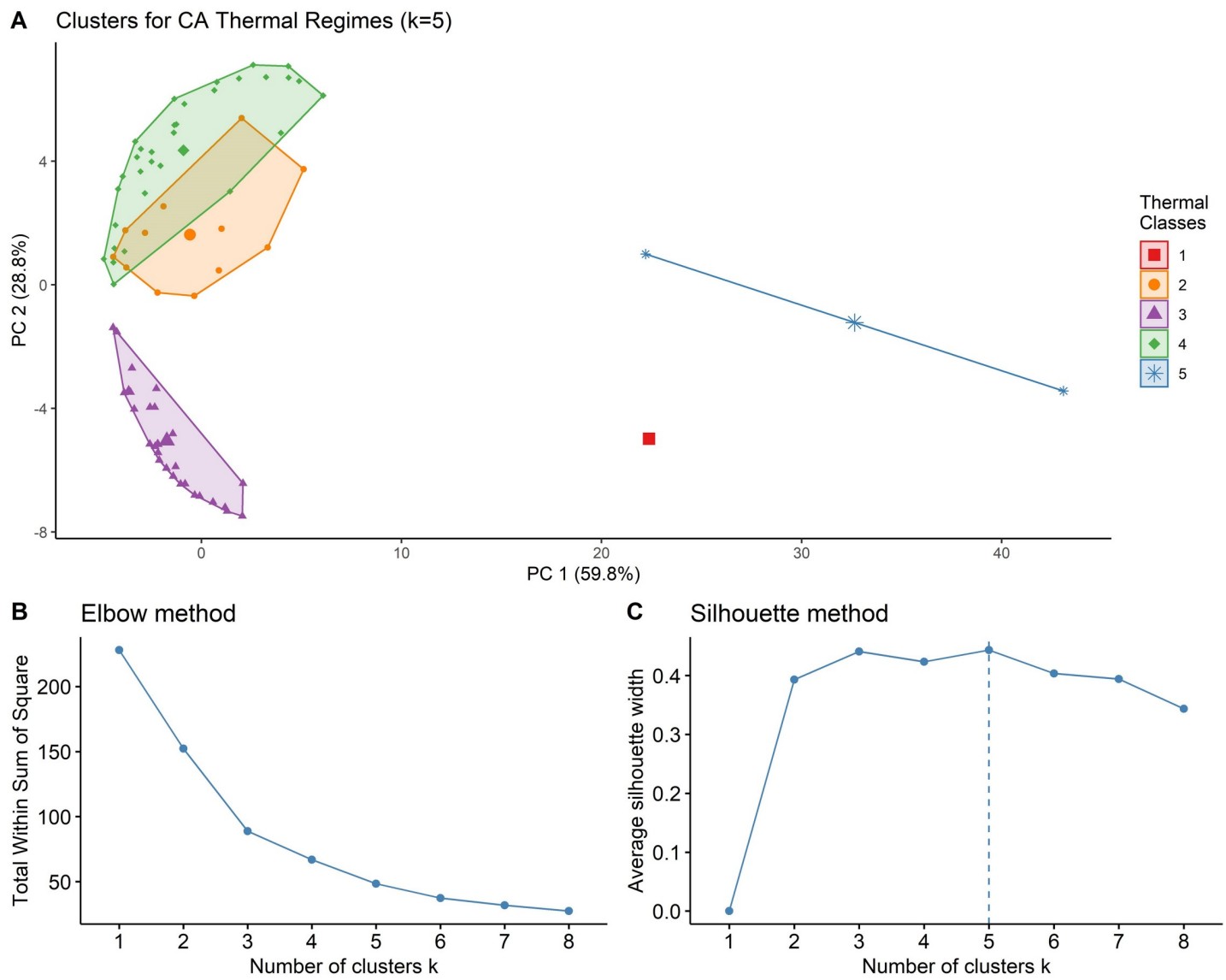

**Fig 3. Results of the clustering analysis.** a) California's thermal regimes were grouped into five clusters, with the centroid of each cluster marked by relatively larger symbols designated for each cluster. The inflection points at k = 5 in the b) elbow and c) silhouette analyses further support the selected groupings.

Thermal regimes were plotted by the five original clusters and named based on patterns in mean annual maximum temperatures ($\bar{T}_{max} > 20$°C was warm; $15$°C $> \bar{T}_{max} > 20$°C cool; $\bar{T}_{max} < 15$°C cold) and their relative annual variability (Fig 4A). Groundwater-fed springs each established their own thermal regimes (stable cold and stable warm), differing in magnitude and timing of annual maximum temperature. The stable warm class was populated by a single site, with the warmest annual maximum and mean temperatures ($\bar{T}_{max} = 28.7$°C, $\bar{T}_{mean}$ of 27.2°C), and the latest day of annual maximum (DOWY = 353, Sept. 19; Fig 4A–4C; Table 1). As this thermal regime described only one site, no assessment could be made of potential variability in this regime. The stable cold regime similarly described a groundwater-fed site, as well as the outlet of Shasta Dam. In contrast to the stable warm regime, the stable cold regime showed the coolest average annual maximum and mean water temperatures ($\bar{T}_{max} = 11.9$°C, $\bar{T}_{mean} = 11.1$°C), and the earliest day of annual maximum (DOWY = 89, Dec. 22; Fig 4A–4C;

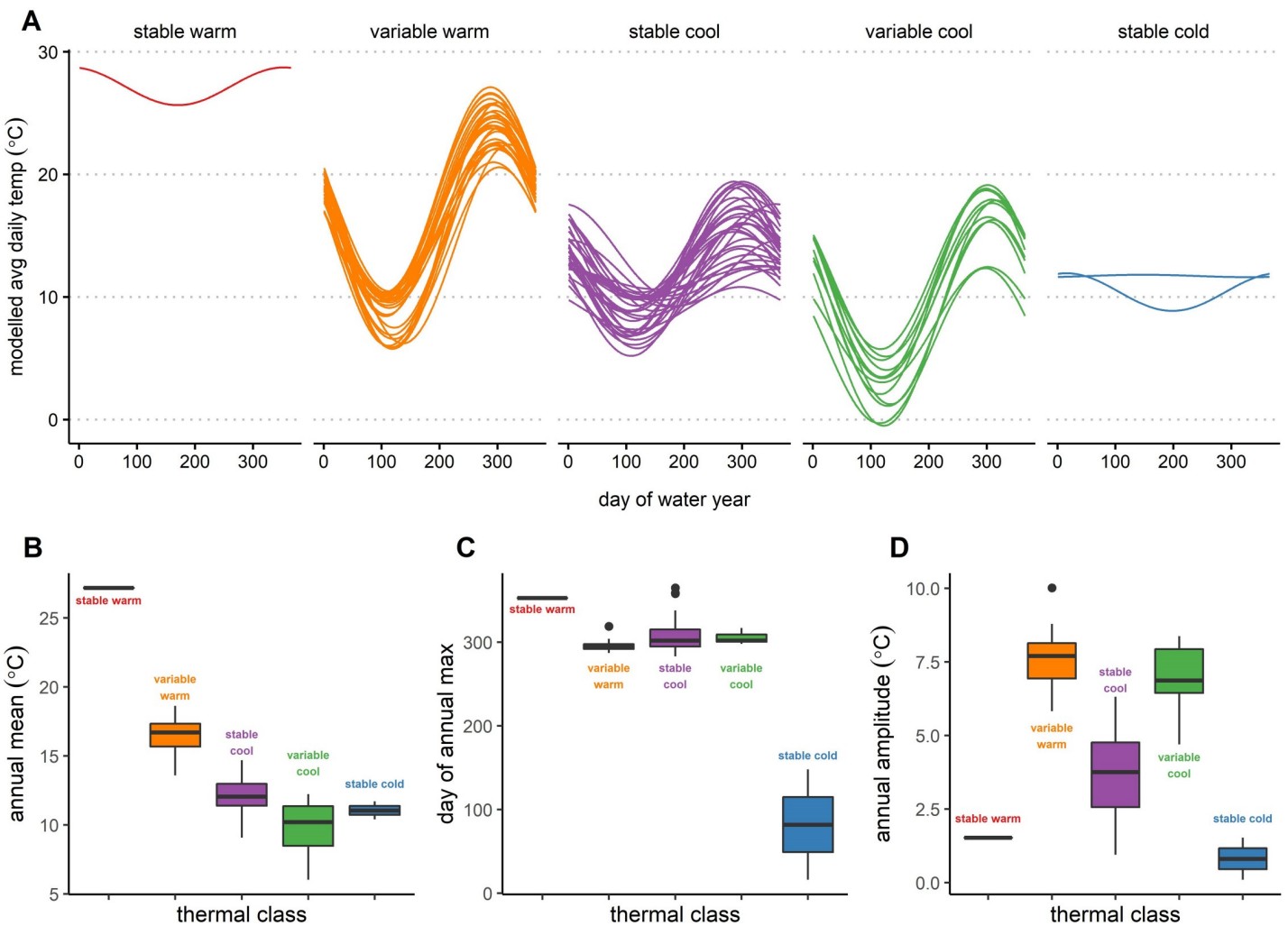

**Fig 4. Thermal regime models and metrics.** a) Classified models and box plots of b) annual mean, c) day of annual maximum, and d) annual amplitude; based on Maheu et al. [31]: Fig 3, with permission. Thermal regimes were characterized based on their mean annual maximum (warm, cool, or cold) and relative annual variability. The number of members for each class (n) is as follows: Stable warm (n = 1), variable warm (n = 30), variable cool (n = 12), stable cool (n = 32), stable cold (n = 2).

Table 1). The two members of this regime showed little variability in annual maximum and mean temperatures, but high variability in the timing of the annual maximum: at Shasta Dam outlet, the annual maximum occurred on DOWY 16 (Oct. 17); at the groundwater spring, it

**Table 1. A summary of thermal regime classes.** Thermal regime class summaries include the number of members (n); average annual maximum and mean water temperatures; and day of annual maximum.

| Thermal regime | n | Average annual maximum (˚C) | Average annual mean (˚C) | Day of annual maximum: DOWY* (Date) |
|---|---|---|---|---|
| Stable warm | 1 | 28.7 | 27.2 | 353 (Sept 19) |
| Variable warm | 30 | 24.0 | 16.4 | 295 (Jul 23) |
| Variable cool | 12 | 16.9 | 9.8 | 305 (Aug 2) |
| Stable cool | 32 | 15.7 | 12.1 | 309 (Aug 6) |
| Stable cold | 2 | 11.9 | 11.1 | 89 (Dec 22) |

*DOWY = Day of Water Year.

occurred on DOWY 148 (Feb. 26, S1 Table). Interestingly, the thermal regime at the Shasta Dam outlet showed the same annual pattern as the stable warm groundwater spring, while the stable cold groundwater spring showed a generally uniform temperature throughout the water year (Fig 4A).

The variable warm regime included 30 sites, with a $\bar{T}_{max}$ = 24.0˚C (DOWY = 295, Jul. 23) and $\bar{T}_{mean}$ = 16.4˚C (Table 1). This thermal regime showed the highest range of annual amplitude (Fig 4D) and second highest annual mean temperature (Fig 4B), both of which illustrated widely variable ranges. Of the classes with multiple members, the variable warm regime had the most consistent day of annual maximum, ranging from DOWY 287–298 (Jul. 15–Jul. 26), with a single site showing its day of annual maximum on DOWY 319 (Aug. 16 at site SCQ, the Tule River at the outlet of Success Dam; S1 Table).

The stable cool regime included 32 sites, with a $\bar{T}_{max}$ = 15.7˚C (DOWY 309, Aug. 6) and $\bar{T}_{mean}$ = 12.1˚C. While the stability was observed in terms of the overall range of annual temperatures across this thermal regime (Fig 4A, Table 1), each classifying metric showed variability across the regime's member sites. The variable cool regime included 12 sites, with a $\bar{T}_{max}$ = 16.9˚C (DOWY 305, Aug. 2) and $\bar{T}_{mean}$ = 9.8˚C (Table 1). In contrast with the stable cool regime, the variable cool regime had a greater variable annual temperature pattern (i.e., the range of temperatures illustrated by the annual trend), but less variable range of annual maximum and mean temperatures, and day of annual maximum.

With the exception of the stable warm regime (which only described a single site), each regime occurred in several hydrologic regions and multiple thermal regimes occurred between the headwaters and mouth of each watershed (Figs 5 and 6, Table 2). Stable cold regimes were found in the North Coast and Sacramento River hydrologic regions; the variable warm and stable cool regimes occurred in the North Coast, Sacramento River, San Francisco Bay, San Joaquin River, and Tulare Lake hydrologic regions. The variable cool regime occurred in the North Coast, North Lahontan, Sacramento River, and San Joaquin River hydrologic regions. The frequency of variable warm sites increased towards inland, southern areas. While the North Coast and Sacramento River hydrologic regions had the same number of stable cool and stable cold sites (n = 12 and n = 1, respectively), the Sacramento River had more variable warm (n = 9 versus n = 3) and fewer variable cool (n = 2 versus n = 4) sites (Table 2). These differences increased in the San Joaquin River hydrologic region, with 15 variable warm sites and 8 stable cold; however, the San Joaquin had more variable cool sites than the Sacramento River (n = 3 versus n = 2).

The relative location of a site to a dam appeared to influence its thermal regime more than its hydrologic region. In general, sites upstream of reservoirs or in unregulated tributaries tended to have a variable cool thermal regime; stable cool regimes often occurred at dam outlets and extended downstream before transitioning to variable warm regimes (Figs 5 and 6). The outlet of Shasta Dam and Success Dam were two exceptions: Shasta Dam produced a stable cold regime at its outlet, while Success Dam (the southern-most site analyzed) produced a variable warm regime (Fig 6A and 6E). Above California's Central Valley rim dams, thermal regimes were exclusively variable cool. In the Central Valley, stable warm regimes generally occurred in the mainstem Sacramento and San Joaquin rivers, despite stable cool regimes in their respective upstream tributaries.

In addition, the length of stream reach affected by an upstream dam varied. Stable cool regimes tended to occur closer to dams, while sites with variable warm regimes were more frequently farther away (Fig 7). Below Shasta, Lewiston, and New Melones dams, stable cool regimes were observed tens of kilometers downstream (Fig 4A–4C). New Melones produced a stable cold thermal regime 83 km below its outlet (site 11303000), the farthest range of

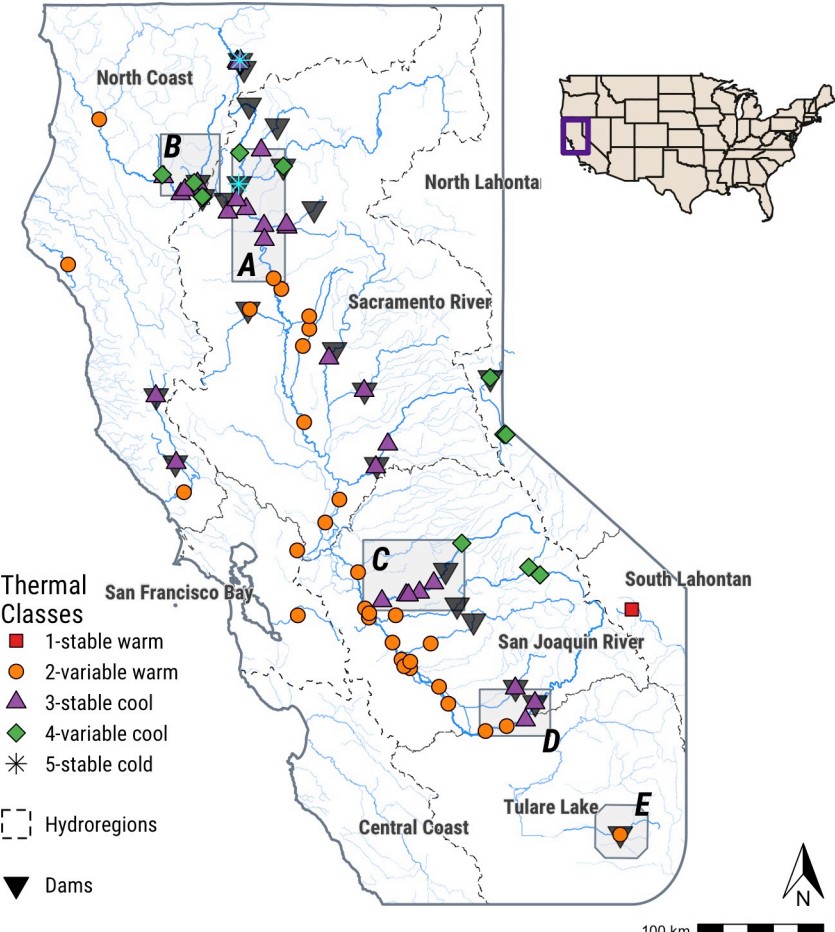

**Fig 5. Map of classified thermal regimes and dams located upstream of study sites in California.** Dotted lines show the borders of California's hydrologic regions as defined by the state Department of Water Resources.

influence observed below any of the dams included in this study (Figs 6C and 7). Success and Black Butte dams produced the shortest distance downstream to variable warm thermal regimes at 0.6 km and 1.9 km, respectively (Fig 5E; Black Butte panel not shown). The remaining dams could maintain stable cool regimes at least 40 km downstream from their outlets before transitioning to variable warm regimes (Fig 7).

## Discussion

### What constitutes a cold-water thermal regime?

In ecological terms, "cool" and "cold" are typically used to classify species based on temperatures that support optimal growth, and are often simplified to static thresholds. These thresholds vary by region: one study identified 10–15°C for cold-water fishes, 21°C for cool, and 30°C for warm in the Great Lakes region [62]; another suggested <20°C, 20–28°C, and >28°C for more general cold, cool, and warm-water optima [63]. Other studies use additional criteria to classify thermal regimes [6,31,41], but still rely on threshold-based definitions for cool and cold. Our results show that, based on criteria developed by Rahel and Olden [63], all but two of California's thermal regimes would be considered cold; yet these supposedly equivalent cold regimes demonstrate a range of ecological performance related to targeted cold-water species

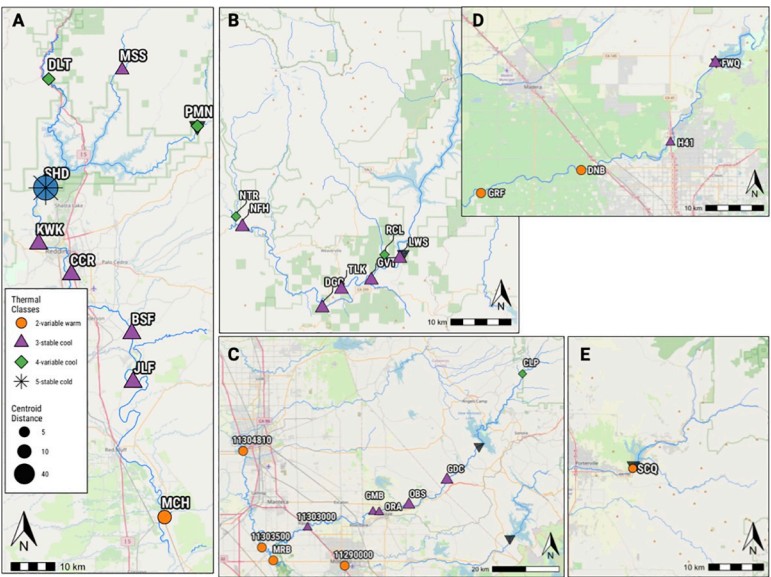

**Fig 6.** Panels of thermal regimes below a) Shasta, b) Lewiston, c) New Melones, d) Friant, and e) Success dams. See Fig 5 for location of each inset map in the study area.

objectives [64]. Regulatory guidance has trended toward more temporally refined thresholds based on target species and their life histories [65]. This analysis used three metrics to characterize annual trends and can be applied as a useful first step to identify desirable areas for cold-water management. Other studies show how additional metrics and short-term variability are useful to assess thermal regimes and their relationship to ecological function [6,41,42], which would be key factors in developing conservation strategies to support process-based thermal regime management. Refined classification, whether based on variability [31], geography [41], or some other feature, are important to distinguish thermal regimes that would otherwise be considered uniformly supportive of cold-water ecosystems using a threshold system.

In addition to developing a classification that captures the range of variability of California's cool- and cold-water regimes, our objective was to provide a classification to support conservation decisions. To this end, the geographic scope of analysis showed important differences from other studies. Interestingly, our results showed more thermal classes for unregulated reaches when California was analyzed as a state as compared to national or regional analyses [31]. Five thermal classes were defined for unregulated sites, compared to three identified by Maheu et al. [31]. Two of those five classes were considered warm (both stable and variable),

**Table 2. Thermal regimes in each hydrologic region.** Hydrologic regions are defined the California Department of Water Resources; the Central Coast, South Coast, and Colorado River are not included as no study sites were located in those regions.

|  | Stable warm | Variable warm | Stable cool | Variable cool | Stable cold |
| --- | --- | --- | --- | --- | --- |
| North Coast | 0 | 3 | 12 | 4 | 1 |
| North Lahontan | 0 | 0 | 0 | 3 | 0 |
| Sacramento River | 0 | 9 | 12 | 2 | 1 |
| San Francisco Bay | 0 | 2 | 0 | 0 | 0 |
| San Joaquin River | 0 | 15 | 8 | 3 | 0 |
| South Lahontan | 1 | 0 | 0 | 0 | 0 |
| Tulare Lake | 0 | 1 | 0 | 0 | 0 |

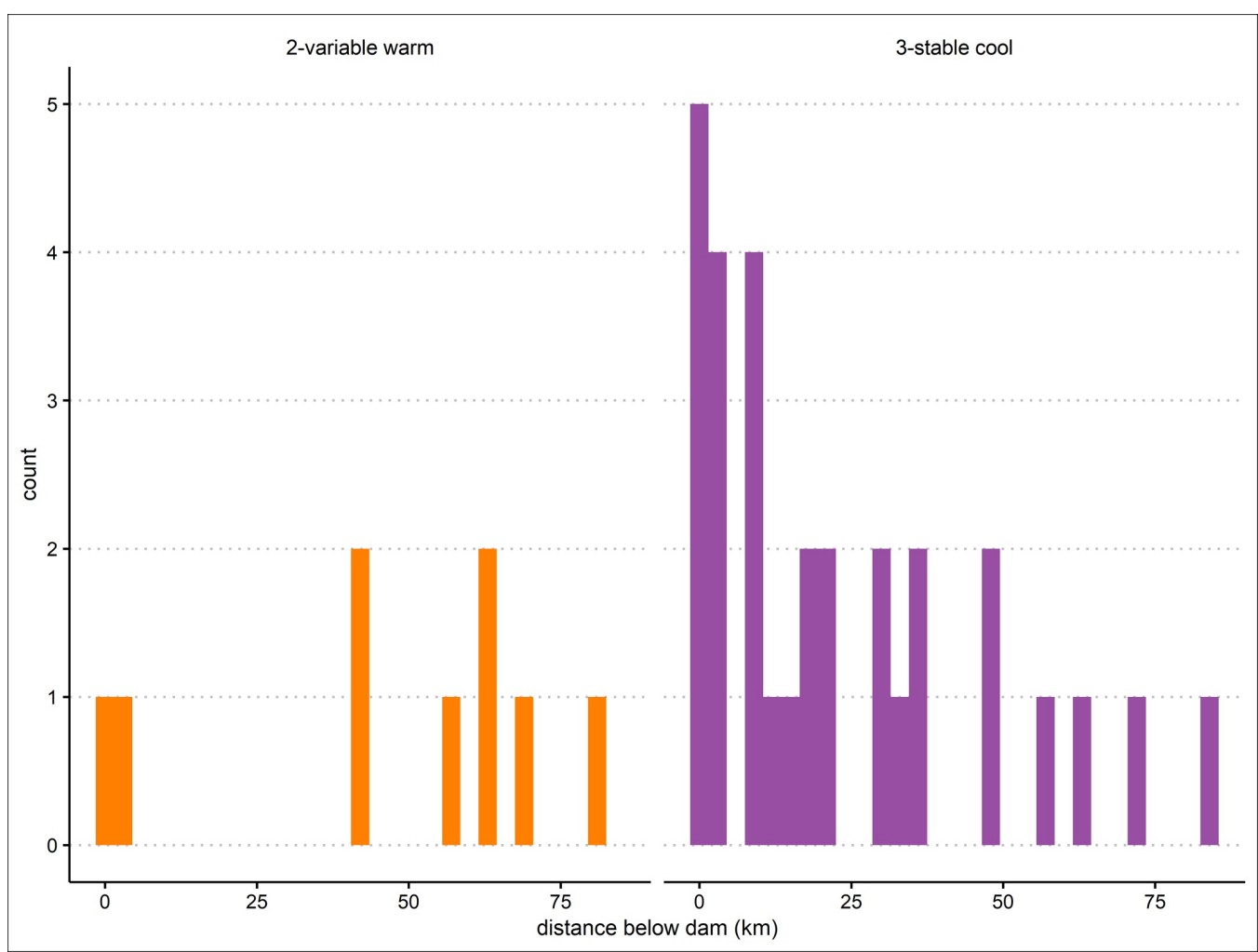

**Fig 7. Histogram of thermal regime location relative to nearest upstream dam.**

while the national analysis resulted in only cool and cold regimes in California. Some of these differences are explained by the inclusion of groundwater-dominated sites in this study, which accounted for two of the five classes. However, other sites that had been classified as cool (or stable) in the national analysis were reclassified as warm (or variable) in our analysis. Isaak et al. [41], which classified thermal regimes in California as part of the western U.S., identified five regime classes in California, though none represented groundwater and showed less diversty in hydrologic regions like the North Coast.

These results show the importance of geographic scope when developing a conservation strategy. Salmonids and other cold-water species have been documented as far south as Mexico [66], indicating that cool- and cold-water regimes extend further south than shown in this study. Though data gaps were not filled in this analysis, recent interpolation methods show promising utility for stream temperature records [67], which could potentially increase the number of sites included in the analysis and expand the geographic scope. Finally, while USGS and CDEC databases have many long-term temperature datasets, and temperature monitoring tends to focus on short-term summer periods, additional data may be available through other public, crowd-sourced sites as more comprehensive temperature data are collected [68].

The five-class system revealed nuanced differences between cool- and cold regimes, and highlighted the importance of groundwater-fed streams to support cold-water conservation. While agglomerative nesting showed comparable statistical strength of classifying California's thermal regimes into three or five classes, fewer classes may be an oversimplification not necessarily useful for management decisions, which typically occur at state or local levels. Research by Null et al. [35] found that thermal regimes of California's western Sierra Nevada rivers did not show the same shifts in desirable cold water habitat as in national-scale studies. Because California rests at the southern edge of the geographic range of many cold-water species [45], has diverse geographic and hydrologic streamscapes [34], and is strongly influenced by dam regulation [36], having high resolution of its thermal regimes and effects of regulation is desirable for conservation planning and investment. Thus, while three classes had slightly stronger statistical support, five classes provided more insightful differences between cool- and cold-water thermal regimes, particularly relative to groundwater and dam releases.

Warm and cold groundwater-fed springs accounted for two of California's five thermal regimes: stable warm and stable cold. Although each class contained a single groundwater-fed site, these regimes illustrated a unique thermal pattern dominated by groundwater-fed spring sources. The stable cold regime, which included both a groundwater-fed spring and the outlet of Shasta Dam, was relatively unstable as indicated by its Jaccard coefficient and large spread of members from the cluster centroid. Additional data describing stable cold sources would improve understanding of this regime by indicating whether it is a stable class with high variability (which would account for the large spread of the initial two members) or better broken down into separate classes, possibly defined by large groundwater-fed springs and dams with large cold-water storage volumes. Despite studies that have classified thermal regimes in California [31,41], none explicitly identified thermal regimes for spring-fed sites in the state. However, the presence of "slightly thermal" groundwater-fed streams in California [50,69] suggests that such a class may be more prevalent than currently known. A separate, stable cold class dominated by releases from reservoirs, though, is unlikely given the dearth of reservoirs in California with the cold-water capacity comparable to Shasta Reservoir.

### Do dams "reset" the longitudinal pattern of a stream's thermal regimes?

Our study shows that dams do not reset thermal regimes: rather, they create discontinuities characterized by artificial regimes. These dynamics may persist for 10s-100s km downstream from a dam's outlet [47,61], and tend to be compounded by multiple upstream dams (S2 Table). Differences between regulated and unregulated thermal regimes (excluding groundwater-dominated regimes) are illustrated by their annual magnitudes and variability. Variable cool regimes occurred exclusively in unregulated reaches, had more variable annual patterns (i.e., warmer annual maximums and cooler minimums), and had more predictable annual means, maximums, and day of annual maximum than stable cool regimes in regulated reaches. As a result of this variability, the sine model was a poorer fit for unregulated sites compared to regulated sites. Regulated thermal regimes showed the opposite trend. Stable cool regimes were strongly influenced by upstream dams and showed less annual variability, but higher variability among the three classifying metrics. Thus, although the overall annual temperature pattern was more stable in regulated reaches, the annual mean, annual maximum, and day of annual maximum varied more within this regime than in variable warm and cool regimes. This variability may relate to storage capacity, operational objectives, geography, or degree of regulation [36,41,48]. Stable cool regimes transitioned to variable warm regimes as the downstream distance from a dam increased. Variable warm regimes were generally at least 40 km downstream of dam outlets and may reflect a transition from regulated influences to dynamic

equilibrium, when stream temperatures are dominated by heat flux due to ambient meteorological conditions [61].

The influence of dams and stable cool thermal regimes was further strengthed once factors like degree of regulation were taken into account. Grantham et al. [36] quantified the degree of regulation for California streams for both individual dams and cumulative number of dams above stream segments. The results for stable cool thermal regimes coincided with reaches that were strongly altered, specifically due to dams (alteration was defined as degree of regulation > = 1; see S2 Table). This extensive downstream effect of dam regulation begs the question of what a thermal regime's natural evolution from headwater to lowland equilibrium may have looked like, and whether dams have eliminated an important transitional reach for temperature-sensitive ecosystem function. Additional analysis of thermal regimes given modeled, unregulated stream temperatures (e.g., [35]) could show the historic fate of thermal regimes over stream reaches currently dominated by dam releases and identify if the transition to equilibrium included regimes similar to those produced downstream of dams, or a more gradual shift in annual mean from variable cool to variable warm regimes.

Most notably, while stable cool regimes successfully mitigated elevated summer stream temperatures, they similarly constrained winter minimum temperatures and maintained artificially warm conditions. Research on the effects of dam regulation on stream temperatures tends to focus on the summer season [47], when elevated stream temperatures may lead to stress or increased mortality of cold-water species [12,45]. Fewer studies have focused on the negative effects of sustained periods of elevated winter temperatures on cold-water species [70]; we are unaware of studies that focus on the potential to replicate colder winter patterns with dam regulation.

## Can dams be managed to replicate desirable cold-water regimes?

In stream reaches that lack a resilience to climate warming, cool- and cold-water habitat may be unachievable through dam regulation. In particular, the stable cool regime may present the greatest challenge to cold water conservation as it generally lacks the cooler winter temperatures of unregulated variable cool regimes. One notable result was the classification of the Shasta Dam outlet (site SHD)–the only reservoir to produce a stable cold thermal regime. At 4.6 million acre feet (MAF), Shasta Lake is California's largest reservoir and maintains its cold pool through cold-water inflows, cooling that occurs during the winter, thermal stratification, and operational decisions [38]. Despite the large capacity of New Melones (2.4 MAF, 4th largest reservoir in California), it, or any other dam included in this analysis, was unable to produce a stable cold regime at its outlet.

This study only considered the effect of the nearest upstream dam to a study site. Many streams have several dams, perhaps with compounded thermal effects [71]. For dams that lack both the capacity to produce a stable or variable cold regimes and lack passage above the dam, these barriers may be insurmountable for species' recovery. While reservoir operation to support cold-water habitat has shown promise [39,72,73], our results suggest that improving passage or dam removal is likely needed to reunite species with the thermal regimes in which their life-history strategies originally evolved. Potential constraints are considerable, though, given the fundamental shift in underlying, unregulated thermal patterns as a result of climate warming, particularly in mid-elevation streams [35].

Finally, despite similarities between groundwater-fed and dam-influenced reaches, the conservation value of these reaches should not be conflated. Streams like the McCloud River, Pit River, Battle Creek, and Big Springs Creek are highly influenced by groundwater-fed springs, and were the only upstream-of-reservoir reaches to replicate the same stable cool thermal

regimes found below reservoirs. The regulated reach downstream of Shasta Dam also illustrates a distinct antinode-node pattern that is characteristic of large-volume, groundwater-fed streams [50,61]. Despite similar thermal regimes, other research has shown how other aspects of these groundwater-dominated streams differ from runoff and regulated reaches [74]. Historically, groundwater-dominated streams have out-produced non-groundwater-dominated streams [74,75], and, for the streams still accessible to fish, are preferentially selected [73,75]. Previous studies have shown even in regions sensitive to climate warming, watersheds with larger flow volumes and groundwater contributions, like California's Feather River, are less vulnerable to climate change [31,35]. Also, some Californian spring-fed streams have novel hydroecological feedbacks that drive their thermal regimes [51], influence reaches tens of kilometers downstream from spring sources [46,50], and support robust ecological productivity and conservation potential [11,74]. Thus, other factors, such as water quality (nutrients), physical habitat, flow regime, and novel ecohydrological feedbacks may still make spring-fed reaches more desirable habitat than regulated reaches despite their similar thermal regimes.

## Conclusions: Thermal regimes and conservation

Conservation planning for cold-water species can be a risky investment in California. The combination of California's location at the southern range of cold-water species, vulnerability to climate warming, and highly regulated streams all pose major challenges. Extinction is likely for most (78%) of California's native salmonids; though altered or degraded thermal regimes are a major stressor, they are not the only limitation [43]. Bold conservation actions are required to reverse the trend towards extinction.

To direct conservation resources effectively to reaches with regulated cold-water regimes in California, strategies should account for extensive regulated influences and capture nuances of highly variable geography and hydrology. Identifying areas where high-quality, cold-water habitats exist and understanding their thermal signatures and function will facilitate prioritization and habitat conservation, in addition to describing the core characteristics necessary for recreating or restoring thermal functionality in other locations. The thermal regime classification developed in this study can be used to identify areas where conservation investment will support the recovery and persistence of valued native species.

## Supporting information

**S1 Table. Model results and metadata.**
(CSV)

**S2 Table. Summary of degree of regulation for regulated sites.** Data for each site's drainage, mean annual runoff, dam storage, degree of regulation, and cumulative degree of regulation were provided by Grantham et al. [36]. Big Springs Dam is a small, privately owned dam; data defining its reservoir's storage capacity was unavailable.
(CSV)

## Acknowledgments

Thank you to Drs. Jay Lund, Steven Sadro, Alexander Forrest, and two anonymous reviewers for their valuable feedback, which greatly improved this manuscript. Thank you to Drs. Audrey Maheu and Ted Grantham for providing data from their research to help contextualize our study.

## Author Contributions

**Conceptualization:** Ann D. Willis, Andrew L. Rypel.

**Data curation:** Ann D. Willis, Ryan A. Peek.

**Formal analysis:** Ann D. Willis, Ryan A. Peek.

**Funding acquisition:** Ann D. Willis.

**Investigation:** Ann D. Willis, Ryan A. Peek.

**Methodology:** Ann D. Willis, Ryan A. Peek.

**Project administration:** Ann D. Willis.

**Software:** Ann D. Willis, Ryan A. Peek.

**Supervision:** Ann D. Willis, Andrew L. Rypel.

**Validation:** Ann D. Willis, Ryan A. Peek.

**Visualization:** Ann D. Willis, Ryan A. Peek.

**Writing – original draft:** Ann D. Willis.

**Writing – review & editing:** Ann D. Willis, Ryan A. Peek, Andrew L. Rypel.

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
