## [Decision Letter · Decision Letter 0]

23 Feb 2021

PONE-D-20-40523

Classifying California's stream thermal regimes for cold-water conservation

PLOS ONE

Dear Dr. Willis,

Thank you for submitting your manuscript to PLOS ONE. After careful consideration, we feel that it has merit but does not fully meet PLOS ONE’s publication criteria as it currently stands. Therefore, we invite you to submit a revised version of the manuscript that addresses the points raised during the review process.

I have completed my evaluation of your manuscript. The reviewers indicated strongs points of the manuscript , and I agree with them. The reviewers and I recommend reconsideration of your manuscript following major revision. I invite you to resubmit your manuscript after addressing the comments below.

We look forward to receiving your revised manuscript.

Kind regards,

João Carlos Nabout

Academic Editor

PLOS ONE

2. We note that Figure 5  in your submission contains map images which may be copyrighted. All PLOS content is published under the Creative Commons Attribution License (CC BY 4.0), which means that the manuscript, images, and Supporting Information files will be freely available online, and any third party is permitted to access, download, copy, distribute, and use these materials in any way, even commercially, with proper attribution. For these reasons, we cannot publish previously copyrighted maps or satellite images created using proprietary data, such as Google software (Google Maps, Street View, and Earth). For more information, see our copyright guidelines: http://journals.plos.org/plosone/s/licenses-and-copyright.

(1) You may seek permission from the original copyright holder of Figure 5 to publish the content specifically under the CC BY 4.0 license. 

Reviewers' comments:

Reviewer's Responses to Questions

**Comments to the Author**

1. Is the manuscript technically sound, and do the data support the conclusions?

Reviewer #1: Partly

Reviewer #2: Yes

2. Has the statistical analysis been performed appropriately and rigorously? 

Reviewer #1: Yes

Reviewer #2: Yes

3. Have the authors made all data underlying the findings in their manuscript fully available?

Reviewer #1: Yes

Reviewer #2: Yes

4. Is the manuscript presented in an intelligible fashion and written in standard English?

Reviewer #1: Yes

Reviewer #2: Yes

5. Review Comments to the Author

Reviewer #1: I enjoyed reading the manuscript, “Classifying California Thermal Regimes” by Willis et al. It is generally well conceived and written. However, I do have some concerns about the use of PCA with a temperature dataset summarized by only three metrics and I suggest the authors also consider a S-type PCA directly on the daily mean temperatures as described below to see what this reveals about temporal dynamics among sites. I also have concerns about the representativeness of the observation dataset for all of California’s streams but this might be rectified by a simple modification of the title as stipulated below. If the authors can address these issues and several additional minor points below, I think the manuscript would be suitable for publication in PLoS.

Lines 47-48. Consider broadening the statement that “nearly 50% of cold-water habitat could be lost due to climate change” to something like “20-90% of cold-water habitat could be lost due to climate change for some species depending on their thermal constraints and landscape resistance to dispersal (LeMoine et al. 2020. Landscape resistance mediates native fish species distribution shifts and vulnerability to climate change in riverscapes. Global Change Biology. doi: 10.1111/gcb. 15281).

Line 51. Consider adding a complimentary reference to 14-McCullough et al., which is Isaak et al. 2018. Global warming of salmon and trout rivers in the Northwestern US. Transactions of the American Fisheries Society 147:566-587 because it describes the actual climate warming trends in the Columbia River and discusses effects on salmon populations, including the mass mortality events that occurred in 2015.

Line 65. Replace “As” with “At” at the beginning of the sentence.

Lines 64-76 paragraph discussing the extent of dam regulation in California. To set the context for later results, I think it would be useful to discuss the range of thermal outcomes that dams may induce on downstream thermal regimes. At opposite extremes, for example, small shallow reservoirs often cause downstream warming whereas large, deep reservoirs with cold hypolimnions cause cooling and dampen variability. Good references in this regard are Olden and Naiman (63) and Maheu (64), which the authors cite later in the discussion.

Line 102 and Figure 5. Water temperature site locations occur primarily at lower elevations in the Central Valley downstream of dams, which makes me wonder if the article’s title shouldn’t be modified to include the word “regulated”? Perhaps “Classifying California’s Regulated Stream Thermal Regimes for Cold-Water Conservation”? No doubt there are many dams in the state and much of the surface hydrology is altered but there are also substantial areas higher in the mountains and in northwestern California where free-flowing streams are common and that don’t appear to be represented in the study’s dataset. To better characterize what is represented I the samples, could the authors modify Table S1 to include descriptor fields such as site elevation, upstream watershed area as a surrogate if mean annual discharge data aren’t available, and the degree of upstream flow regulation?

Lines 107-108. It’s very common to have missing data values in water temperature time series. Did those occur here, and if so, how were they treated?

Line 119. Seems awkward to say “methods developed in (31)”, here and elsewhere. Maybe instead say “methods developed by Maheu et al. (31)”

Line 138. It’s unclear where the principle analysis comes from as this is the first time it is mentioned. Please provide more information here on the PCA. It was only apparent later in the results section that the three summary metrics (annual mean, amplitude, and phase) were the subject of the analysis. It also seemed strange to do a PCA on so few metrics, as the analysis is usually done when there are numerous metrics to search for and summarize commonality and orthogonality among them. With only three metrics considered, the PCA plot in Figure 3 is almost identical to what a simple scatterplot of the phase metric vs. mean and/or amplitude would show and it’s hard to justify the additional analytical complexity. To pull more out of this dataset with PCA, the authors may want to consider a S-Type PCA wherein the analysis is run directly on the daily mean temperatures from the 77 sites. It’s simple to do and will highlight individual sites that behave as outliers and those which conform to broader group dynamics. A recent example of S-mode PCA applied to stream temperature time series is provided by Isaak et al. 2018. Principal components of thermal regimes in mountain river networks. Hydrology and Earth System Sciences 22:6225-6240.

Line 161. Insert “showed” into phrase “Modeling results a reasonable…”

Reviewer #2: This study uses a clustering method to enable classification of thermal regimes of around 70 sites across California, roles of dams in influencing thermal regimes and explored the possibility of using dams as regulators of thermal regimes. Although the information presented here is a useful for conservation planning and prioritization, the novelty of the questions or the method discussed here is hard to decipher. Also, the study did not attempt to investigate the different thermal regimes in detail such as correlate the presence of different thermal regimes to the flow regimes, microclimate, surface-subsurface interactions etc and primarily attributed varying thermal regimes to presence of dams. Hence, major revision is suggested in order for authors to address these concerns.

Comments:

Line 33-36: Don't understand the context/relevance of this sentence.

Line 46-51: Why are you just considering dams here? Addition of heated effluents (i.e. water pollution), among other factors, also causes warming of river reaches.

Line 107: Did your dataset have missing values? please mention how you dealt with missing values

Line 121: The classification of thermal regimes used 3 parameters for clustering namely, annual mean, amplitude, and phase. Why were other parameters such as timing, frequency (in lines with important parameters for flow regime) not used?

Line 290-293: How did you account for other sources of warming/cooling? In other words, how could you be sure that the observed regimes changes were solely due to dams?

Line 293-294: Sentence not clear, please reframe

Line 332-334: Considering that you used relatively long-term data (>5yrs), how did you account for shifts in thermal regimes and thermal trends?

Line 357: How does your classification spatially compare with other thermal classifications done for California as well as other national level classifications?

Figures: Please improve figure 5 (map), zoom in further to the study area

General: Please review text to eliminate grammatical errors and typos and to improve sentence framing.

6. PLOS authors have the option to publish the peer review history of their article (what does this mean?). If published, this will include your full peer review and any attached files.

Reviewer #1: No

Reviewer #2: No

---

## [Author Response · Author response to Decision Letter 0]

20 Apr 2021

Academic Editor’s comments:

Thank you for pointing out our need to align our manuscript with PLOS ONE’s style requirements, including those for file naming. We have reviewed the templates and adjusted our manuscript styles and file names accordingly.

2. We note that Figure 5 in your submission contains map images which may be copyrighted. All PLOS content is published under the Creative Commons Attribution License (CC BY 4.0), which means that the manuscript, images, and Supporting Information files will be freely available online, and any third party is permitted to access, download, copy, distribute, and use these materials in any way, even commercially, with proper attribution. For these reasons, we cannot publish previously copyrighted maps or satellite images created using proprietary data, such as Google software (Google Maps, Street View, and Earth). For more information, see our copyright guidelines: http://journals.plos.org/plosone/s/licenses-and-copyright.

Thank you for drawing attention to the copyrighted material underlying the paneled maps in Figure 5. We recreated all mapped figures using Open Street Map layers that adhere to terms of the Creative Commons Attribution License. 

Reviewer 1’s comments:

General comment: I enjoyed reading the manuscript, “Classifying California Thermal Regimes” by Willis et al. It is generally well conceived and written. However, I do have some concerns about the use of PCA with a temperature dataset summarized by only three metrics and I suggest the authors also consider a S-type PCA directly on the daily mean temperatures as described below to see what this reveals about temporal dynamics among sites. I also have concerns about the representativeness of the observation dataset for all of California’s streams but this might be rectified by a simple modification of the title as stipulated below. If the authors can address these issues and several additional minor points below, I think the manuscript would be suitable for publication in PLoS.

Thank you for your review. We appreciate your interest, and thought that your questions helped us develop several areas in our study where we had not clearly explained our motivations or objectives. We have provided more extensive answers to the individual comments listed below, including those related to our use of the PCA and the broader context of our study beyond regulated thermal regimes. In addition, we further developed the work around related dam regulation to thermal regimes, and appreciated the comment as the additional work helped strengthen our findings. We hope that they, combined with the revisions we made in the manuscript, address the concerns raised in this review. 

Lines 47-48. Consider broadening the statement that “nearly 50% of cold-water habitat could be lost due to climate change” to something like “20-90% of cold-water habitat could be lost due to climate change for some species depending on their thermal constraints and landscape resistance to dispersal (LeMoine et al. 2020. Landscape resistance mediates native fish species distribution shifts and vulnerability to climate change in riverscapes. Global Change Biology. doi: 10.1111/gcb. 15281).

Thank you for providing an updated citation! Our understanding of the results in LeMoine et al. (2020) are that the 50% estimate is generally consistent with the Eaton and Scheller (1996) findings, with the broader potential losses occurring for sensitive species like slimy sculpin. The results that seemed most directly related to the suggested revision included the findings of 50% extirpation probabilities for bull trout and slimy sculpin (z-scores 0.79 and 0.23), with much higher probabilities occurring at z-scores > 1 or < 0 (results summarized in LeMoine et al. 2020 Figure 4). Given this useful and more detailed study, we revised our statement to read: “Across the United States, projections show nearly 50% of cold-water habitat could be lost due to climate change (10, 20), though this decline varies widely depending on species, their thermal constraints, and landscape resistance to dispersal (20).”

Line 51. Consider adding a complimentary reference to 14-McCullough et al., which is Isaak et al. 2018. Global warming of salmon and trout rivers in the Northwestern US. Transactions of the American Fisheries Society 147:566-587 because it describes the actual climate warming trends in the Columbia River and discusses effects on salmon populations, including the mass mortality events that occurred in 2015.

Thank you for suggesting a complimentary reference to 14-McCullough et al. We have added it to our manuscript.

Line 65. Replace “As” with “At” at the beginning of the sentence.

Thank you for suggesting the clarifying revision. We replaced “As” with “At” at the beginning of the sentence.

Lines 64-76 paragraph discussing the extent of dam regulation in California. To set the context for later results, I think it would be useful to discuss the range of thermal outcomes that dams may induce on downstream thermal regimes. At opposite extremes, for example, small shallow reservoirs often cause downstream warming whereas large, deep reservoirs with cold hypolimnions cause cooling and dampen variability. Good references in this regard are Olden and Naiman (63) and Maheu (64), which the authors cite later in the discussion.

Thank you for the suggestion. We agree that this would be a good place to set the context for later results, and appreciate the suggested framing. We added the line “Where studies have explore the effects of dams, the results suggest that they produce variable thermal regimes depending on size, the ability to selectively withdraw water depending on temperature (i.e., whether a dam possesses the necessary infrastructure to adjust the depth at which it draws water for releases), and operational objective(s) (43, 44); these regulated thermal regimes may or may not align with existing, unregulated regimes.” 

Line 102 and Figure 5. Water temperature site locations occur primarily at lower elevations in the Central Valley downstream of dams, which makes me wonder if the article’s title shouldn’t be modified to include the word “regulated”? Perhaps “Classifying California’s Regulated Stream Thermal Regimes for Cold-Water Conservation”? No doubt there are many dams in the state and much of the surface hydrology is altered but there are also substantial areas higher in the mountains and in northwestern California where free-flowing streams are common and that don’t appear to be represented in the study’s dataset. To better characterize what is represented [I] the samples, could the authors modify Table S1 to include descriptor fields such as site elevation, upstream watershed area as a surrogate if mean annual discharge data aren’t available, and the degree of upstream flow regulation?

Thank you for the comment, as it has helped us think more carefully about the main points of our paper and expand more on some ideas that we may not have explored thoroughly in the original draft. Because the method we are using requires a relatively long dataset, relatively few sites met the analysis criteria. Of these sites, 14 were unregulated, and included two groundwater-dominant sites. Though this seems like a relatively small number, it is comparable to other studies using methods requiring long-term data sets for California. Maheu et al. (2016), which focused on using long-term datasets to classify the thermal regimes of unregulated sites, included 11 sites in California in runoff-dominated watersheds. We are unaware of other studies that take the same approach of using long-term data to develop a single annual trend that include more sites for California. We also think it is important to keep our classification broad to explore how unregulated site classification compared to other studies once we included regulated sites and narrowed the geographic scope to only California. Finally, while many of the sites are below dams, we observe that there is a limit to the extent of dam influence, particularly in the Central Valley. Thus, while the flows may be regulated, the thermal regimes adopt an equilibrium signature that seems independent of regulation. We have expanded our discussion of these points in the discussion sections of the manuscript to clarify the relevance of the unregulated classification results in our study. We have also added language to the first section of our discussion that specifically identifies this limitation and area for future work.

We agree that Table S1 could be modified with additional descriptor fields to be clearer about descriptor metrics that are relevant to the thermal regime classification. We found it cleaner to make additional tables, and have submitted Tables S2 and S3. Table S2 provides a summary of regulated and unregulated sites. Table S3 provides more details for the regulated sites, including upstream watershed area, mean annual discharge, degree of regulation, and cumulative degree of regulation for each site (based on the results published in Grantham et al. 2014). 

Lines 107-108. It’s very common to have missing data values in water temperature time series. Did those occur here, and if so, how were they treated?

Thank you for the question. Indeed, we found many years that were incomplete. Because we wanted to focus on full, empirical datasets, and because the methodology we were following meant we would average all daily average temperatures for a given day into a single, annual trend, we eliminated any sites that had fewer than 8 daily average observations for any given day for the full period of record. This greatly reduced the number of potential sites that met the criteria of full datasets for a minimum of 8 years of daily data. We did not interpolate or fill data gaps in our study, which may have increased the number of sites we could have included, but also left us concerned that we might introduce bias into the annual trends of sites for which we filled data gaps. We added this clarifying language into the manuscript: “All data were reviewed to remove flagged data (per USGS and CDEC standards) and obvious outliers; the remaining years with a minimum of 8 daily average temperature observations for each day were used in the study.”

Line 119. Seems awkward to say “methods developed in (31)”, here and elsewhere. Maybe instead say “methods developed by Maheu et al. (31)”

Thank you for the observation and suggested revision. We agree that our original language was awkward, and have revised three instances where we referred to Maheu et al. only by its reference number. We confirmed with the PLOS-ONE editorial support that this was change was in line with the journal formatting, as we had previously misunderstood the guidelines (which is what had prompted the original, awkward language). Thanks again for the note. See lines 131-32, 148, and 233 for the changes.

Line 138. It’s unclear where the principle analysis comes from as this is the first time it is mentioned. Please provide more information here on the PCA. It was only apparent later in the results section that the three summary metrics (annual mean, amplitude, and phase) were the subject of the analysis. It also seemed strange to do a PCA on so few metrics, as the analysis is usually done when there are numerous metrics to search for and summarize commonality and orthogonality among them. With only three metrics considered, the PCA plot in Figure 3 is almost identical to what a simple scatterplot of the phase metric vs. mean and/or amplitude would show and it’s hard to justify the additional analytical complexity. To pull more out of this dataset with PCA, the authors may want to consider a S-Type PCA wherein the analysis is run directly on the daily mean temperatures from the 77 sites. It’s simple to do and will highlight individual sites that behave as outliers and those which conform to broader group dynamics. A recent example of S-mode PCA applied to stream temperature time series is provided by Isaak et al. 2018. Principal components of thermal regimes in mountain river networks. Hydrology and Earth System Sciences 22:6225-6240.

Thank you for the comment. There were a few reasons why we choose to include a PCA analysis, which we will explain here and expand on in our manuscript as well. The primary advantage of using a PCA was to easily explore the variation explained by a given metric, normalized against the variation of other metrics. Though we focused on three metrics, we preferred to use PCA to visualize the clusters on a uniform, two-dimensional scale. In addition to more clearly emphasize the effect of each metric on the overall classification, the PCA illustrated the relationship between cluster strength and regulation at each site. We expanded our explanation of our motivations and objectives in the methods section.

In considering this comment, we also completed an S-Type PCA to see if the results varied from our original analysis. The results of the S-Type PCA were consistent with our original findings, which encouraged us. But as the findings were not substantively different from our original analysis, we opted to keep our initial approach for our methods. Thank you again for the comment, and for the suggestion of an alternative that helped us explore our results from another perspective.

Line 161. Insert “showed” into phrase “Modeling results a reasonable…”

Thank you for catching the error. We revised the sentence to read, “Modelling results showed a reasonable sine curve fit for all sites included in the study…”

Reviewer 2 comments:

General comment: This study uses a clustering method to enable classification of thermal regimes of around 70 sites across California, roles of dams in influencing thermal regimes and explored the possibility of using dams as regulators of thermal regimes. Although the information presented here is a useful for conservation planning and prioritization, the novelty of the questions or the method discussed here is hard to decipher. Also, the study did not attempt to investigate the different thermal regimes in detail such as correlate the presence of different thermal regimes to the flow regimes, microclimate, surface-subsurface interactions etc and primarily attributed varying thermal regimes to presence of dams. Hence, major revision is suggested in order for authors to address these concerns.

Thank you for your review. We appreciate the insightful questions and specific issues you identified, and are pleased to have an opportunity to further develop our study. The novelty of our work is shown in three aspects: exploring the relationship of stream thermal regimes in the context of regulated versus unregulated reaches, the role of groundwater-dominated streams in creating distinct classes of unregulated thermal regimes, the oversight of poorly replicated winter thermal patterns by dams, and how the spatial scope of a study influences actionable science outcomes. We have expanded our discussion of each of these topics to more clearly show our study’s contribution to the broader field of stream temperature research.

The focus of this study was to explore thermal regimes given a few easily obtainable metrics so that it could be widely replicated. While flow regimes, microclimate, surface-subsurface interactions, etc, can influence discrete temperatures, such data is rarely available on the scale necessary to explore landscape thermal regimes in depth and were beyond the scope of this study. Our results also suggest the metrics we included were well-suited to identify and distinguish different cold-water thermal regimes in California. We find it meaningful that, despite the different drivers and buffers in thermal regimes, all unregulated regimes classified the same, with the exception of groundwater-dominated systems. However, we expanded our exploration of the role of dams and their ability to account for our results by incorporating data from Grantham et al. (2014) about watershed area, mean annual runoff, and degree of regulation (both as a result of dams located directly upstream of each site and the cumulative degree of regulation of all upstream dams). 

Comments:

Line 33-36: Don't understand the context/relevance of this sentence.

Thank you for your comment. The sentence introduces the application of water temperature analysis for regulation and management and the different ways it has been approached. As our study explores the application of thermal regimes as the framework for guiding conservation, we wanted to acknowledge another approach that was historically more common (i.e., identifying target temperature thresholds to manage cold water for aquatic ecosystems) before focusing on the body of research that explores temperature using a thermal regime framework. 

Line 46-51: Why are you just considering dams here? Addition of heated effluents (i.e. water pollution), among other factors, also causes warming of river reaches.

Thank you for your question. We are particularly interested in dams for two reasons: first, because dams have widely been shown to disrupt temperature patterns without much research quantifying their long-term thermal regimes and how they compare to unregulated thermal regimes; second, because much work has been dedicated in the engineering literature to suggest that dams can be operated to achieve desirable thermal regimes without actually testing this hypothesis against the extensive work that primarily exists in the ecological research community around thermal regimes necessary to support aquatic ecosystems. In California, these objectives combine with its geographic vulnerability to climate change to present a powerful confluence of ecosystem goals with severe consequences if they are not achieved. 

We acknowledge and agree that there are other potential sources of heating (heated effluent, degraded channel and riparian zones, stream diversions, etc.). We focused on one – groundwater dominated systems – because these were the only reaches where notable variability that was unrelated to dam regulated was illustrated by our results. We expected to see a broader range of regimes if other factors produced important variability. Our results did not support the hypothesis that other factors were necessary to understand the difference in thermal regimes on a landscape scale. Thus, given that our results did not show evidence of a strong influence on thermal regimes unrelated to regulation or groundwater-dominated streams, we did not extend our analysis to include specific alternative factors. 

Line 107: Did your dataset have missing values? [p]lease mention how you dealt with missing values

Thank you for the question. Indeed, we found many years that were incomplete. Because we wanted to focus on full, empirical datasets, and because the methodology we were following meant we would average all daily average temperatures for a given day into a single, annual trend, we eliminated any sites that had fewer than 8 daily average observations for any given day for the full period of record. This greatly reduced the number of potential sites that met the criteria of full datasets for a minimum of 8 years of daily data. We did not interpolate or fill data gaps in our study, which may have increased the number of sites we could have included, but also left us concerned that we might introduce bias into the annual trends of sites for which we filled data gaps. We added this clarifying language into the manuscript: “All data were reviewed to remove flagged data (per USGS and CDEC standards) and obvious outliers; the remaining years with a minimum of 8 daily average temperature observations for each day were used in the study.”

Line 121: The classification of thermal regimes used 3 parameters for clustering namely, annual mean, amplitude, and phase. Why were other parameters such as timing, frequency (in lines with important parameters for flow regime) not used?

We wanted to adhere as closely as possible to the methods used in Maheu et al. (2016) so that we could understand potential differences between our results and previous results by changing only one variable: including regulated reaches. Thus, we wanted to introduce as little change to the analytic methods as possible. Though an exploration of timing and frequency were beyond the scope of this study, we are continuing our research by exploring how those factors fit into thermal regime management, too. We added language to the discussion to emphasize the need for additional research in this area.

Line 290-293: How did you account for other sources of warming/cooling? In other words, how could you be sure that the observed regimes changes were solely due to dams?

Thank you for the question, as it gets to the crux of our findings related to dam influences on thermal regimes. While we didn’t explicitly account for other sources of warming/cooling, we would expect our results to show more variability in regimes if those sources were dominant drivers/buffers. The shift in thermal regime from variable cool to stable cool were consistent with dams occurring in between sites, and occurred in each stream included in the analysis for the entire geographic region of California. While there are other sources of warming/cooling, it seems unlikely that a unique combination of those feedbacks occur throughout the study area, independent of the dams, which would explain the consistent shift across dam sites. We have expanded the discussion in the section related to dams’ influences on thermal regimes to address this issue.

Line 293-294: Sentence not clear, please reframe

Thank you for the comment. We have revised the sentence into two, which now read as: “Specifically, we found that many different thermal regimes could be defined as “cold,” though these regimes vary depending on what metrics are used. Therefore, when conservation efforts focus on managing cold-water ecosystems, multiple metrics should be considered to replicate an effective “cold” thermal regime at the local or regional scale.”

Line 332-334: Considering that you used relatively long-term data (>5yrs), how did you account for shifts in thermal regimes and thermal trends?

Thank you for the question. We did not explicitly account for shifts in thermal regimes and thermal trends. However, we are aware that given climate change and California’s geographic vulnerability to stream warming, this could be a potential issue. To explore whether members showed evidence of shifting thermal regimes, we used the PCA results to identify strong and weak members. We have added language to the discussion specifying the limits of this classification method, particularly if used to implement actionable science outcomes for conservation. 

Line 357: How does your classification spatially compare with other thermal classifications done for California as well as other national level classifications?

Thank you for the question. We have added language to our discussion that puts our results in the context of other thermal classifications. 

Figures: Please improve figure 5 (map), zoom in further to the study area

Thank you for your comment. We have revised figure 5 to more clearly illustrate the study area and results, and have created a companion figure that focuses on the inset panels identified in the map. Please see figures 5 and 6. 

General: Please review text to eliminate grammatical errors and typos and to improve sentence framing.

Thank you for the comment. We have reviewed the text to eliminate grammatical errors/typos and reframe sentences.

---

## [Decision Letter · Decision Letter 1]

17 Jun 2021

PONE-D-20-40523R1

Classifying California's stream thermal regimes for cold-water conservation

PLOS ONE

Dear Dr. Willis,

Thank you for submitting your manuscript to PLOS ONE. After careful consideration, we feel that it has merit but does not fully meet PLOS ONE’s publication criteria as it currently stands. Therefore, we invite you to submit a revised version of the manuscript that addresses the points raised during the review process.

I have completed my evaluation of your manuscript. The reviewer see clear improvement in the revision and only ask minor adjustments (see below). I invite you to resubmit your manuscript after addressing the comments below.

We look forward to receiving your revised manuscript.

Kind regards,

João Carlos Nabout

Academic Editor

PLOS ONE

Journal Requirements:

Additional Editor Comments (if provided):

Reviewers' comments:

Reviewer's Responses to Questions

**Comments to the Author**

1. If the authors have adequately addressed your comments raised in a previous round of review and you feel that this manuscript is now acceptable for publication, you may indicate that here to bypass the “Comments to the Author” section, enter your conflict of interest statement in the “Confidential to Editor” section, and submit your "Accept" recommendation.

Reviewer #1: (No Response)

Reviewer #2: All comments have been addressed

2. Is the manuscript technically sound, and do the data support the conclusions?

Reviewer #1: Partly

Reviewer #2: Yes

3. Has the statistical analysis been performed appropriately and rigorously? 

Reviewer #1: Yes

Reviewer #2: Yes

4. Have the authors made all data underlying the findings in their manuscript fully available?

Reviewer #1: Yes

Reviewer #2: Yes

5. Is the manuscript presented in an intelligible fashion and written in standard English?

Reviewer #1: Yes

Reviewer #2: Yes

6. Review Comments to the Author

Reviewer #1: Manuscript Number: PONE-D-20-40523R1

Manuscript Title: Classifying California's stream thermal regimes for cold-water conservation

I reread the revised manuscript and think that the author’s have done a generally good job responding to concerns raised in my original review. If the author’s can address several additional items listed below, I think the paper could be made acceptable for publication on PONE. Most of the items are relatively minor, with the exception of revising the discussion section as noted below.

1. The Abstract needs revision. I’d suggest adding a few sentences between lines 25 and 26 to describe the dataset and analytical procedures. Otherwise, the conclusion statements starting on line 26 have no foundation.

2. Line 87. Sentence starts awkwardly, “Other, more data and computationally…” and could use revision.

3. Line 91 states “This study develops a classification framework…for rapid identification of stream reaches likely to sustain cool- and cold-water regimes.” The phrase “likely to sustain” implies to me that a temporal trend analysis will be done such that reaches which will remain cold in the future are being identified, despite climate change or other factors that may cause warming. I’d slightly rephrase this by deleting “likely to…” from the sentence since the analysis of regimes here is based on classifying discriminating characteristics for an eight-year snapshot of time.

4. Line 128. Do the a, b, and n coefficients correspond to the mean, amplitude, and phase? This isn’t clear from the text or the associated figure.

5. Lines 187-188. This sentence belongs in the methods section. The standard name for this type of graph is an ordination plot I believe.

6. Line 193. If PC2 is most strongly correlated with phase, and the stable cold category of stream reaches a peak earlier than all the other classes (Fig 4c), why does this class plot intermediately along the PC2 axis (Fig 3) rather than at one of the extremes?

Discussion section

1. One limitation of the Maheu three parameter approach is that it ignores short-term variability (e.g., daily cycles and weekly variation) because it’s smoothing the annual cycle with a sine wave fit. In Isaak et al.’s 2020 classification of western U.S. stream thermal regimes based on dozens of metrics (reference 60 cited by the authors), that short-term variability was the primary determinant of PC2 (as was also the case in Rivers-Moore et al. 2013 multi-metric classification of South African streams; reference 46 cited by the authors) and others studying thermal regimes, as described in the discussion section of the Isaak paper, have argued that short-term thermal variation has particular ecological importance. In the discussion section of the present manuscript, it would be useful for the authors to elaborate on potential tradeoffs associated with using different metric sets for regime description and classification.

2. Lines 335-338. There are numerous papers that have already developed metrics to describe and explore stream thermal regimes based on frequency, rate of change, duration, magnitude, etc. some of which should be cited here (e.g., Steel et al. 2017 (reference 8 cited by authors in earlier context); Rivers-Moore et al. 2013 (reference 46 cited by authors in earlier context).

3. Paragraph lines 339-349. I think this paragraph needs significant revision because the potential already exists to mine information from a much larger database than the USGS & CDEC gage datasets that form the basis of the author’s analysis. The publicly available NorWeST database (https://www.fs.fed.us/rm/boise/AWAE/projects/NorWeST.html) contains stream temperature records for 3,681 unique sites in California as part of a much larger west-wide database. The dataset was published by Isaak et al. 2017 (Water Resources Research 53:9181-9205). Granted, most of the NorWeST records consist of short summer-only monitoring records but many do not and the records are easily sortable to extract those with more comprehensive records for regime analysis.

Also relevant to this paragraph of the discussion is the utility of modern imputation techniques for filling gaps in temperature records. In my experience, these work remarkably well with stream temperature records at both regulated and unregulated sites due to the strong temporal synchrony among sites, especially when the sites are part of dense monitoring networks as is the case here. The recent paper by Johnson et al. 2021 (Heed the data gap: guidelines for using incomplete datasets in annual stream temperature analyses. Ecological Indicators 122:107229) highlights the application of the imputation techniques developed by Josse et al. (2012. Handling missing values in exploratory multivariate data analysis methods, Journal of the Société Francaise de Statistique, 153, 79–99; and Josse and Husson 2016. MissMDA: a package for handling missing values in multivariate data analysis, Journal of Statistical Software 70: 1–31) to stream temperature records.

4. The discussion section as a whole at 11 pages is quite long compared to the overall 25 pages of text. I’d recommend looking for opportunities to streamline so that the strengths of the paper are highlighted while more speculative elements of the discussion are shortened or eliminated.

Reviewer #2: Thank you for revising your manuscript based on the comments. The revisions have made the manuscript clearer and more robust. In general, specific responses to the reviewer comments and associated revisions in the manuscript seem satisfactory. I do have a few minor comments to those below. All in all, this manuscript is an important contribution to the field of river temperature research and suitable for publication in PlosOne.

Minor comments:

Comment on Your response to Reviewer 2’ comment on Line 46-51:

Your explanation to the comment clarifies the rationale behind focusing on regulated reaches and the inclusion (or exclusion) of other factors. However, this rationale does not come across as clearly in the manuscript. I suggest to make this rationale more explicit in the introduction. The rationale in the introduction should also mention the novelty or research gaps that you are addressing (such as including in the paragraph starting line 91).

Comment on Your response to Reviewer 2’ comment on Line 107:

Your response clarifies how you dealt with missing values. Although you have included a clarificatory line in the methods pertaining to this, I think it should be mentioned explicitly that data gaps/missing values were used as it as and not filled.

Lines 220-259: Including a table showing different thermal classes and their characteristics (mean, max, min, n, CV, DOWY etc) would be useful for better comprehension and for reducing the amount of text in these paras.

Figure 4: Please also include the ‘n’ for each class within in the figure/legend/figure title.

There are still plenty of grammatical errors in the manuscript. Please correct them. Correcting some typos below:

Line 27: Groundwater streams are not a class of thermal regimes. They may exhibit a certain class of thermal regimes.

Line 34: worth “the” investment

Line 38: Replace “whereas” with “while”

Line 82: explore”d”

Line 146: "example" instead of examples

Line 151: Principal Components Analysis (PCA)

Line 395: “Some of these differences are” instead of “Some of this difference is”

Line 401: “importance of” instead of “important of”

Line 552: review”ers”

7. PLOS authors have the option to publish the peer review history of their article (what does this mean?). If published, this will include your full peer review and any attached files.

Reviewer #1: No

Reviewer #2: No

---

## [Author Response · Author response to Decision Letter 1]

27 Jul 2021

Reviewer #1: Manuscript Number: PONE-D-20-40523R1

Manuscript Title: Classifying California's stream thermal regimes for cold-water conservation

I reread the revised manuscript and think that the author’s have done a generally good job responding to concerns raised in my original review. If the author’s can address several additional items listed below, I think the paper could be made acceptable for publication on PONE. Most of the items are relatively minor, with the exception of revising the discussion section as noted below.

1. The Abstract needs revision. I’d suggest adding a few sentences between lines 25 and 26 to describe the dataset and analytical procedures. Otherwise, the conclusion statements starting on line 26 have no foundation.

Thank you for the comment. We have revised the abstract to describe the dataset and general analytical procedures. We revised the abstract to include the following:

“We used publicly available, long-term (> 8 years) stream temperature data from 77 sites across California to model their thermal regimes, calculate three temperature metrics, and use the metrics to classify each regime with an agglomerative nesting algorithm. Then, we assessed the variation in each class and considered underlying physical or anthropogenic factors that could explain differences between classes. Finally, we considered how different classes might fit existing criteria for cool- or cold-water thermal regimes, and how those differences complicate efforts to manage stream temperature through regulation.”

2. Line 87. Sentence starts awkwardly, “Other, more data and computationally…” and could use revision.

Thank you for the suggestion. We have revised the sentence to read as follows:

“Other approaches that require less data (< 5 years) or are computationally efficient bring considerable uncertainty in the results (10, 45, 46).”

3. Line 91 states “This study develops a classification framework…for rapid identification of stream reaches likely to sustain cool- and cold-water regimes.” The phrase “likely to sustain” implies to me that a temporal trend analysis will be done such that reaches which will remain cold in the future are being identified, despite climate change or other factors that may cause warming. I’d slightly rephrase this by deleting “likely to…” from the sentence since the analysis of regimes here is based on classifying discriminating characteristics for an eight-year snapshot of time.

Thank you for the comment (and suggestion for further analysis!). We have revised the sentence to replace “likely to sustain viable” with “with”.

4. Line 128. Do the a, b, and n coefficients correspond to the mean, amplitude, and phase? This isn’t clear from the text or the associated figure.

Thank you for the observation. Yes, coefficients a, b, and no correspond to annual mean, annual amplitude, and phase. We have clarified this in the manuscript with the following:

“…where Tw is water temperature, n is the day of water year, and a, b, and no are coefficients that correspond to annual mean, annual amplitude, and phase (Fig 1). Coefficients a, b, and no were optimized using least square regression.”

5. Lines 187-188. This sentence belongs in the methods section. The standard name for this type of graph is an ordination plot I believe.

Thank you for the suggestion. We removed those lines from the results, and revised the methods (line 160) to read: “We visualized the distribution of the clusters with an ordination plot of the first two principal components from the analysis, grouped by cluster.”

6. Line 193. If PC2 is most strongly correlated with phase, and the stable cold category of stream reaches a peak earlier than all the other classes (Fig 4c), why does this class plot intermediately along the PC2 axis (Fig 3) rather than at one of the extremes?

Thank you for the question. While PC2 is correlated with phase, but is not completely driven by phase; thus we would not expect to see the class plot entirely at one of the extremes. Nevertheless, when we look at the position of each member in that group on the plot, they fall strongly along the PC2 axis, but in opposite directions. Thus, the centroid is located along the PC2 zero axis, but the members themselves illustrate a more extreme relationship.

Discussion section

General comment: as we have completed a major revision of the discussion, we have not listed every instance of revised language as much of the feedback has been incorporated throughout the discussion. We have provided examples of lines where we specifically incorporated comments into our revisions.

1. One limitation of the Maheu three parameter approach is that it ignores short-term variability (e.g., daily cycles and weekly variation) because it’s smoothing the annual cycle with a sine wave fit. In Isaak et al.’s 2020 classification of western U.S. stream thermal regimes based on dozens of metrics (reference 60 cited by the authors), that short-term variability was the primary determinant of PC2 (as was also the case in Rivers-Moore et al. 2013 multi-metric classification of South African streams; reference 46 cited by the authors) and others studying thermal regimes, as described in the discussion section of the Isaak paper, have argued that short-term thermal variation has particular ecological importance. In the discussion section of the present manuscript, it would be useful for the authors to elaborate on potential tradeoffs associated with using different metric sets for regime description and classification.

Our method can be used as a rough cut to identify areas where conservation investment should be prioritized, particularly where desirable thermal regimes are independent of regulation. Once those areas have been identified, additional metrics such as those described in Isaak et al. and Rivers-Moore et al. can be used to understand the key metrics whose short-term variability has particular ecological importance, and that can be supported through process-based thermal regime management. We have added language to our discussion to specifically identify these limitations of our study (example, lines 334-339). We similarly address some of the comments regarding the range of metrics that have been developed, availability of long-term datasets, and interpolation methods (comments 2 & 3 below). We have condensed our discussion of these issues in the discussion section to focus more on the strengths of the findings, as per comment #4 below.

2. Lines 335-338. There are numerous papers that have already developed metrics to describe and explore stream thermal regimes based on frequency, rate of change, duration, magnitude, etc. some of which should be cited here (e.g., Steel et al. 2017 (reference 8 cited by authors in earlier context); Rivers-Moore et al. 2013 (reference 46 cited by authors in earlier context).

Thank you for the comment. We have added language to the discussion to address this issue, cited the recommended studies (example, lines 334-339), and eliminated the speculative language around more research needed in this area from our discussion.

3. Paragraph lines 339-349. I think this paragraph needs significant revision because the potential already exists to mine information from a much larger database than the USGS & CDEC gage datasets that form the basis of the author’s analysis. The publicly available NorWeST database (https://www.fs.fed.us/rm/boise/AWAE/projects/NorWeST.html) contains stream temperature records for 3,681 unique sites in California as part of a much larger west-wide database. The dataset was published by Isaak et al. 2017 (Water Resources Research 53:9181-9205). Granted, most of the NorWeST records consist of short summer-only monitoring records but many do not and the records are easily sortable to extract those with more comprehensive records for regime analysis.

Also relevant to this paragraph of the discussion is the utility of modern imputation techniques for filling gaps in temperature records. In my experience, these work remarkably well with stream temperature records at both regulated and unregulated sites due to the strong temporal synchrony among sites, especially when the sites are part of dense monitoring networks as is the case here. The recent paper by Johnson et al. 2021 (Heed the data gap: guidelines for using incomplete datasets in annual stream temperature analyses. Ecological Indicators 122:107229) highlights the application of the imputation techniques developed by Josse et al. (2012. Handling missing values in exploratory multivariate data analysis methods, Journal of the Société Francaise de Statistique, 153, 79–99; and Josse and Husson 2016. MissMDA: a package for handling missing values in multivariate data analysis, Journal of Statistical Software 70: 1–31) to stream temperature records.

Thank you for the comments. Regarding the observation of data available from NorWeST, we admit that we considered using this resource, but two issues persuaded us to limit our data queries to USGS and CDEC. First, our general impression from previous exploration with this database was that the majority of temperature records were short-term, summer-only monitoring. Second, the workflow of finding the longer-term records from NorWeST was not intuitive to us. Admittedly, this may be something that becomes less of an issue as we continue to explore and gain experience with this database. But at the time of this analysis, our lack of familiarity and previous experience with the NorWeST database prevented us from fully exploring its potential to supplement our study sites. Regardless, we have noted the database as a potential source for other studies in our manuscript (example: lines 362-365), and look forward to working more with it. Thank you for the suggestion and encouragement.

We have added language to our discussion to address the potential of filling datagaps using the methods identified in the suggested citations (example: lines 360-362). Thank you very much for the recommended sources.

4. The discussion section as a whole at 11 pages is quite long compared to the overall 25 pages of text. I’d recommend looking for opportunities to streamline so that the strengths of the paper are highlighted while more speculative elements of the discussion are shortened or eliminated.

Thank you for the comment. Characterizing some of the discussion as speculative was an incredibly helpful way of framing areas that crowded the stronger areas, and helped us quickly evaluate whether a statement or paragraph should be shortened or eliminated. We have made a major revision to our discussion to focus on the main areas of insight related to the results, and removed the more speculative observations. The revised discussion focuses on the definition of cold when referring to thermal regimes and the implications of alternative methods, and the role of dams in replicating/managing these regimes. We removed several paragraphs from the beginning of the discussion, which were mainly preamble or included statements that were repeated and explored in the context of the results later in that section. We also shorted our discussion of the methods, and included those points in relation to the other discussion topics. The remaining discussion allowed us to focus on the stronger elements of the paper, and retain additional discussion from the previous revision that expanded on dam regulation.

Reviewer #2: Thank you for revising your manuscript based on the comments. The revisions have made the manuscript clearer and more robust. In general, specific responses to the reviewer comments and associated revisions in the manuscript seem satisfactory. I do have a few minor comments to those below. All in all, this manuscript is an important contribution to the field of river temperature research and suitable for publication in PlosOne.

Minor comments:

Comment on Your response to Reviewer 2’ comment on Line 46-51:

Your explanation to the comment clarifies the rationale behind focusing on regulated reaches and the inclusion (or exclusion) of other factors. However, this rationale does not come across as clearly in the manuscript. I suggest to make this rationale more explicit in the introduction. The rationale in the introduction should also mention the novelty or research gaps that you are addressing (such as including in the paragraph starting line 91).

Thank you for your comment and advice about the clarity of our explanation in our first response to reviewer comments. We have revised the introduction to include the clearer language in our earlier response and identify the research gaps we are addressing. 

Comment on Your response to Reviewer 2’ comment on Line 107:

Your response clarifies how you dealt with missing values. Although you have included a clarificatory line in the methods pertaining to this, I think it should be mentioned explicitly that data gaps/missing values were used as it as and not filled.

Thank you for the suggestion. We have added the following line to the methods: “Any data gaps or missing values were not filled; the data was used as is.”

Lines 220-259: Including a table showing different thermal classes and their characteristics (mean, max, min, n, CV, DOWY etc) would be useful for better comprehension and for reducing the amount of text in these paras.

Thank you for the suggestion. We have added a table to the main manuscript summarizing the number of members (n); average annual maximum and mean water temperatures; and day of annual maximum. We have also revised the text to remove some instances where we explicitly catalog the results, with the exception of statements where the results were necessary for clarity.

Figure 4: Please also include the ‘n’ for each class within in the figure/legend/figure title.

Thank you for the suggestion. The caption clarifies which portion of the figure indicates the ‘n’ for each class with the following language: “The number of members for each class (n) is as follows: stable warm (n = 1), variable warm (n = 30), variable cool (n = 12), stable cool (n = 32), stable cold (n = 2).”

There are still plenty of grammatical errors in the manuscript. Please correct them. Correcting some typos below:

Line 27: Groundwater streams are not a class of thermal regimes. They may exhibit a certain class of thermal regimes.

Line 34: worth “the” investment

Line 38: Replace “whereas” with “while”

Line 82: explore”d”

Line 146: "example" instead of examples

Line 151: Principal Components Analysis (PCA)

Line 395: “Some of these differences are” instead of “Some of this difference is”

Line 401: “importance of” instead of “important of”

Line 552: review”ers”

Thank you for your careful reading. We have corrected the errors and typos that were listed, as well as others found during our revision.

---

## [Editor Report · Decision Letter 2]

4 Aug 2021

Classifying California's stream thermal regimes for cold-water conservation

PONE-D-20-40523R2

Dear Dr. Willis,

We’re pleased to inform you that your manuscript has been judged scientifically suitable for publication and will be formally accepted for publication once it meets all outstanding technical requirements.

Kind regards,

João Carlos Nabout

Academic Editor

PLOS ONE
---

## [Editor Report · Acceptance letter]

10 Aug 2021

PONE-D-20-40523R2 

Classifying California’s stream thermal regimes for cold-water conservation 

Dear Dr. Willis:

I'm pleased to inform you that your manuscript has been deemed suitable for publication in PLOS ONE. Congratulations! Your manuscript is now with our production department. 

Kind regards, 

on behalf of

Dr. João Carlos Nabout 

Academic Editor

PLOS ONE